# An auditory brain-computer interface based on selective attention to multiple tone streams

Simon Kojima[1]*, Shin'ichiro Kanoh[1,2]

**1** Graduate School of Engineering and Science, Shibaura Institute of Technology, Koto-ku, Tokyo, Japan, **2** College of Engineering, Shibaura Institute of Technology, Koto-ku, Tokyo, Japan

* nb21106@shibaura-it.ac.jp

**Data Availability Statement:** All relevant data and source code used for analysis within the paper and its Supporting information files are publicly available from the Harvard Dataverse repository (https://doi.org/10.7910/DVN/MQOVEY).

## Abstract

In this study, we attempted to improve brain-computer interface (BCI) systems by means of auditory stream segregation in which alternately presented tones are perceived as sequences of various different tones (streams). A 3-class BCI using three tone sequences, which were perceived as three different tone streams, was investigated and evaluated. Each presented musical tone was generated by a software synthesizer. Eleven subjects took part in the experiment. Stimuli were presented to each user's right ear. Subjects were requested to attend to one of three streams and to count the number of target stimuli in the attended stream. In addition, 64-channel electroencephalogram (EEG) and two-channel electrooculogram (EOG) signals were recorded from participants with a sampling frequency of 1000 Hz. The measured EEG data were classified based on Riemannian geometry to detect the object of the subject's selective attention. P300 activity was elicited by the target stimuli in the segregated tone streams. In five out of eleven subjects, P300 activity was elicited only by the target stimuli included in the attended stream. In a 10-fold cross validation test, a classification accuracy over 80% for five subjects and over 75% for nine subjects was achieved. For subjects whose accuracy was lower than 75%, either the P300 was also elicited for nonattended streams or the amplitude of P300 was small. It was concluded that the number of selected BCI systems based on auditory stream segregation can be increased to three classes, and these classes can be detected by a single ear without the aid of any visual modality.

## Introduction

The brain-computer interface (BCI) has been studied as an effective communication tool for patients who have neuromuscular disorders, such as amyotrophic lateral sclerosis (ALS) [1, 2]. In this system, the user's electrophysiological signal from the brain (e.g., electroencephalogram (EEG)) is measured and analyzed to detect their intention and to control external devices. To date, many types of BCI systems have been reported.

In many BCI systems, one way to realize it is to detect event-related potentials (ERPs). ERPs are a time-locked response by the brain that occur at a fixed time after a particular external or internal event [3].

**Funding:** This work was supported by JSPS KAKENHI (Grant Number JP23K11811 to S. Kanoh) The URL of JSPS KAKENHI is as follows https://www.jsps.go.jp/english/e-grants/grants09_kiban.html. The funders had no role in study design, data collection and analysis, decision to publish, or preparation of the manuscript.

**Competing interests:** The authors have declared that no competing interests exist.

The oddball paradigm is the specific set of circumstances for eliciting the P300 ERP [4]. In the oddball paradigm, standard stimuli are presented repeatedly and deviant stimuli are presented randomly at a low probability. When a subject attends to the deviant stimulus, P300 which is a largely positive response is elicited approximately 300ms after the onset of the deviant stimulus. P300 can be composed into two major components, a frontally maximal P3a and a parietally maximal P3b [5]. It was reported the P3b component is observed for targets that are infrequent but are in some sense expected or awaited, whereas the frontal P3 wave is elicited by stimuli that are truly unexpected or surprising [5]. The amplitude of P300 changes over the midline electrodes (Fz, Cz and Pz) that increases from the frontal to the parietal electrode [6]. The *context-updating model* is known as the account of the functional role of P300. According to the model, as stimuli are presented and evaluated, the degree to which the events are consistent with the current model of the context is assessed. When an event violates the expectations dictated by the model, and when the violation requires the model to be rivised (i.e., *context updating*), a P300 is elicited [4].

## Auditory BCIs

P300-based BCI systems reported to date mainly use a visual modality. One of the renowned visual P300 BCIs, a P300 speller, was attempted by Farwell and Donchin [7]. In this system, six-by-six matrices containing alphabet letters are presented, one of these rows or columns flashes in random order, and subjects are requested to attend to a target letter. The elicited P300 component was analyzed and detected to determine the target letter. Many studies have been done on P300 speller [8–10]. However, systems using visual stimuli occupy the user's sight, and visually impaired people cannot use the system.

Another way to create a BCI is to use auditory stimuli. One of the renowned earlier studies of auditory BCIs was attempted by Hill et al. [11]. Two oddball sequences with different inter-stimulus intervals (ISIs) were presented to each ear of a participant. Subjects were requested to pay attention to one of the two sequences. Recorded EEG signals were classified by a support vector machine (SVM) to detect the user's intention.

Schreuder et al. proposed an auditory BCI that could be used to select one out of eight sequences, each of which was presented from eight speakers surrounding the subject [12]. Eight different tone stimuli were presented from these speakers with a fixed interval in random order, and each speaker provided a fixed tone. Subjects were requested to pay attention to one of these sound sources. The stimuli from the attended sound source elicited P300 activity, and machine learning was used to detect the sound source that users attended.

Auditory BCI systems using auditory steady-state response (ASSR) have also been reported. Lopez et al. [13] and Kim et al. [14] proposed ASSR-based auditory BCIs. In these systems, two modulated (e.g., amplitude modulation) tone with different frequencies were presented to each participant's ear. By attending to or ignoring these stimuli, the power of the alpha-band EEG and ASSR were changed, and the user's selection could be detected.

P300 spellers using auditory stimuli were tested by Furdea et al. [15]. A five-by-five matrix containing the letters of the alphabet was visually presented, and auditory stimuli of spoken numbers were assigned to each row and column. The visually presented matrix was only used for support and did not flash. Subjects were requested to attend to the target spoken number. The output character was decided in two-steps; the row was chosen in the first step, and the column was chosen in the second. Markovinović et al. proposed a similar auditory speller with the help of a convolutional neural network (CNN) [16].

In most of the proposed auditory BCI systems, users were requested to pay attention to one out of multiple tone sequences that were presented from different audio sources (e.g., left or

right, select one from spatially located multiple audio sources), and these systems did not use the properties of tones (e.g., frequency, intensity, and timbre) to create a variety of stimuli. The number of audio sources should be increased if the number of selections is increased. However, more audio sources would decrease the participant's ability to detect the target source. Additionally, it is not practical to place many speakers around a subject.

The authors proposed a 2-class auditory BCI system based on auditory stream segregation [17]. Auditory stream segregation is one of the auditory illusions that is studied in the field of psychoacoustics [18–20]. When two kinds of tones (A and B) are presented alternately (ABABAB. . ..), such a tone sequence is perceived as two different auditory streams (AAAA. . .. and BBBB. . ..). When two tone sequences have a larger gap in the frequency and shorter time intervals, the tones are perceived more clearly. In this BCI system, two oddball sequences consisting of tone bursts with two different frequency ranges were presented alternately to the subject's right ear with a short time interval so that they would be perceived as two segregated tone streams. Subjects were requested to pay attention to one of two oddball sequences. P300 activity and mismatch negativity (MMN) were elicited by target stimuli in the attended oddball tone sequence, and recorded EEG signals could be classified to detect which oddball sequence the subject attended. A similar system based on our research was also used by another researcher, Pokorny et al, on minimally conscious patients [21].

Current auditory BCIs based on stream segregation only offers binary selection; therefore, selection capability needs to be increased for the system to be used as a practical BCI. If the number of choices is increased, the number of presenting streams is increased; however, it would be more difficult to segregate presented tone sequences as multiple streams.

Thus, in this study, instead of pure tones used in the previous study, musical tones with complex harmonics were adopted to facilitate discrimination among these streams. Since musical tones with complex harmonics could have more information that would allow users to group similar tones easier than pure tones, it is expected that discrimination among tone streams would be easier. We tested a 3-class auditory BCI system based on auditory stream segregation in which tone sequences consist of musical tones. Fig 1 shows a conceptual diagram of this system. Three tone streams consist of musical tones are presented to the subject, and the subject paid attention to one of the streams. The attention to the streams was detected by analyzing and classifying the subjects' EEG.

## Materials and methods

Musical tones generated by a digital auditory workstation (Cakewalk by BandLab, BandLab Technologies, Singapore) were used as auditory stimuli. Piano tones (Grand Piano 1 SE) included in the MIDI sound source (SampleTank3, IK multimedia Production, Italy) were used. A digital signal processor (System3, Tucker-Davis Technologies, USA) and headphones (HDA200, Sennheiser) were used to present these tones to the participants. Timing of presented tones was generated by Arduino UNO (Arduino, USA).

Fig 2 denotes the auditory paradigm used in the experiment, and Table 1 shows the frequency of each tone. Each stream n ($n = 1, 2, 3$) consists of standard tone $S_n$ and deviant tone $D_n$. The probabilities of target and nontarget stimuli were 0.1 and 0.9, respectively. The duration of each tone was 150 ms, and the stimulus onset asynchrony (SOA) was set to 180 ms. Auditory stimuli were presented only to the right ear of each participant.

The 64-channel EEG signals (Fp1, Fp2, AF7, AF3, AFz, AF4, AF8, F7, F5, F3, F1, Fz, F2, F4, F6, F8, FT9, FT7, FC5, FC3, FC1, FCz, FC2, FC4, FC6, FT8, FT10, T7, C5, C3, C1, Cz, C2, C4, C6, T8, TP9, TP7, CP5, CP3, CP1, CPz, CP2, CP4, CP6, TP8, TP10, P7, P5, P3, P1, Pz, P2, P4, P6, P8, PO7, PO3, POz, PO4, PO8, O1, Oz, and O2) were measured by Ag-AgCl electrodes

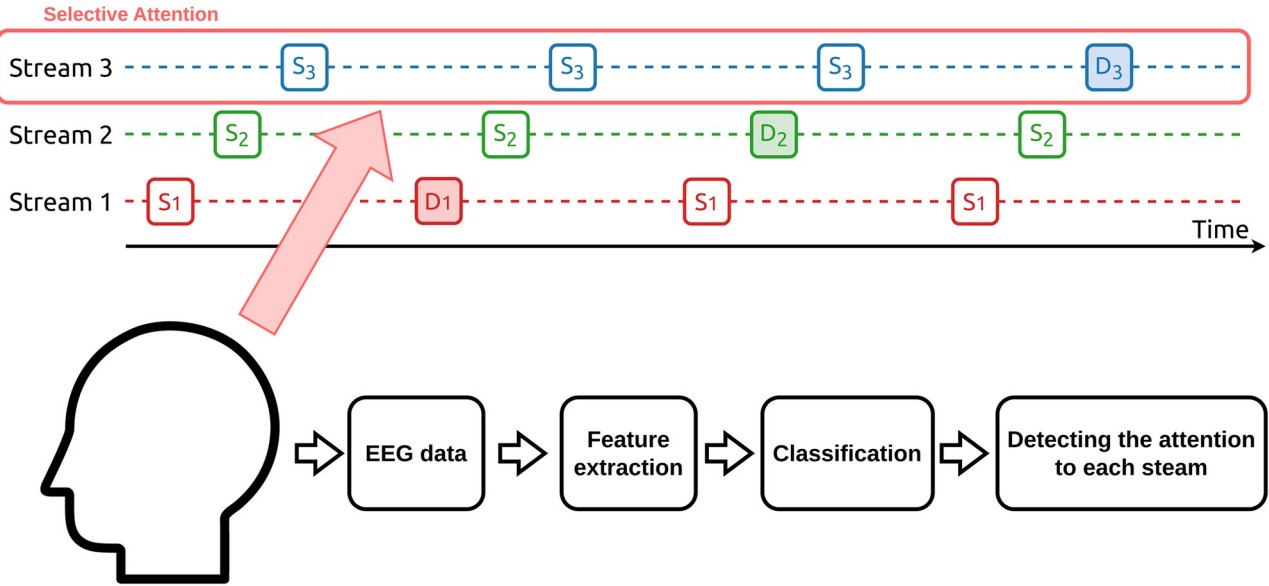

**Fig 1. The conceptual diagram of this system.**

(EASYCAP GmbH, Germany). See Fig 3 for EEG montage. Brain Amp DC (Brain Products GmbH, Germany) and MR plus (Brain Products GmbH, Germany) were used for data acquisition. Reference and ground electrodes were placed on the right and left earlobes, respectively. Vertical and horizontal electrooculogram (EOG) signals were also recorded. Amplified signals were bandpass filtered at 0.1 Hz to 100 Hz and recorded with a sampling frequency of 1000 Hz.

Ten male and one female (aged between 22–23 years) subjects participated in the experiment. This study protocol was approved by the Review Board on Bioengineering Research Ethics of Shibaura Institute of Technology and was conducted in accordance with the Declaration of Helsinki. Before the experiment, subjects were given information orally and in writing, and written informed consent was obtained from all subjects. Subjects were recruited from July 18, 2023, to November 27, 2023.

Fig 4 shows time chart of the session. Firstly, all participants had a familiarization block to learn the paradigm. Each experiment consists of two task blocks. Three runs were conducted in each task block. Each measurement took five minutes to complete. Subjects were requested to count the number of target stimuli in Streams 1, 2, and 3 on the 1st, 2nd, and 3rd measurements, respectively. The same block was repeated twice, and participants rested between blocks.

Data analysis was performed by MATLAB (2021b, MathWorks). Recorded signals were bandpass filtered at 0.1 Hz to 40 Hz (zero-phase 2nd-order Butterworth IIR filter, slope 24

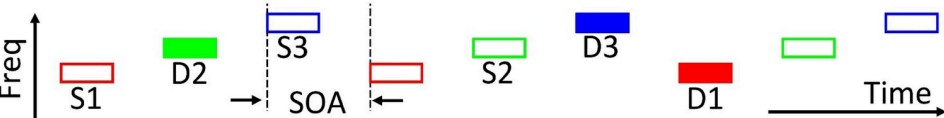

**Fig 2. Time chart of the presented tone sequence.**

**Table 1. Frequencies of tones.**

| Stream 1 | $S_1$ | C3 (131 Hz) |
|---|---|---|
| | $D_1$ | G3 (196 Hz) |
| Stream 2 | $S_2$ | D5 (587 Hz) |
| | $D_2$ | A5 (880 Hz) |
| Stream 3 | $S_3$ | E7 (2637 Hz) |
| | $D_3$ | B7 (3951 Hz) |

dB/octave). It is suggested to use a filter with a slope of between 12 and 24 dB/octave for ERP analysis to avoid distortions produced by filtering [5]. In this study, the filter with a slope of 24 dB/octave was selected to minimize distortion while removing noises. Responses to each target stimulus in the range of −100 ms to 500 ms from onset were extracted, and the mean amplitude at baseline (−50 ms to 0 ms) was subtracted from each response. Epochs in which the amplitude exceeded ±100$\mu$V on EEG recordings and ±500$\mu$V on EOG recordings were excluded from further analysis. Epoched data were averaged over trials in attended and nonattended streams. Scalp topographies were plotted by EEGLAB (v2021.0) [22]. Responses to the target stimuli corresponding to the attended stream and nonattended stream were tested by Student's t-test ($p < 0.01$). Let $D_t \in R^{N_{ch} \times N_t \times N_{et}}$ as responses to the target stimuli corresponding to the attended stream, and let $D_{nt1} \in R^{N_{ch} \times N_t \times N_{e1}}$ and $D_{nt2} \in R^{N_{ch} \times N_t \times N_{e2}}$ ($N_{ch}$ is the number of channels, $N_t$ is the number of time samples and $N_{ex}$ is the number of epochs) as responses to the target stimuli corresponding to the nonattended stream, since there were one attended stream and two nonattended streams. Responses to the stimuli corresponding to the nonattended streams were concatenated over epochs ($D_{nt} \in R^{N_{ch} \times N_t \times (N_{e1} + N_{e2})}$). For each channel and time sample, the significant difference between $D_t$ and $D_{nt}$ was tested.

## Pattern classification

Fig 5 shows the flowchart of the classification pipeline. Pattern classification was performed in Python. Recorded signals were bandpass filtered at 1 Hz to 40 Hz (zero-phase 2nd-order Butterworth IIR filter, slope 24 dB/octave). All the responses to the target stimulus included in both attended and nonattended streams were extracted. Each epoch was extracted in the range of −100 ms to 500 ms from the onset, and the mean amplitude at baseline (−50 ms to 0 ms) was subtracted from each response. Epochs that exceeded ±100$\mu$V on EEG and ±500$\mu$V on EOG recordings were rejected and excluded from further analysis. After that, xDAWN filters [23, 24] (number of components = 3) were extracted for both attended and nonattended epochs to enhance ERP responses and reduce the dimension of the feature vector. The algorithm xDAWN was used to obtain the spatial filter that estimates the evoked subspace that contains most ERP responses and improves the signal-to-signal + noise ratio (SSNR) of ERP responses. xDAWN filter $\mathbf{u}$, which maximizes SSNR can be estimated as following [24]. Let $X \in R^{N_t \times N_s}$, where $N_t$ is the number of time samples and $N_s$ is the number of channels, as recorded EEG data, $P_1 \in R^{N_1 \times N_s}$, where $N_1$ is the number of time samples of the epoch, as the ERP response. SSNR $\rho(\mathbf{u})$ is defined as $\frac{\mathbf{u}^T \Sigma_1 \mathbf{u}}{\mathbf{u}^T \Sigma_x \mathbf{u}}$ with $\Sigma_1 = E[P_1^T D_1^T D_1 P_1]$ and $\Sigma_x = E[X^T X]$, where $D_1 \in R^{N_t \times N_1}$ is Toeplitz matrix with the first element of each row is 1 for stimulus onset and 0 for otherwise. Finally, estimated xDAWN spatial filter is $\mathbf{u} = \arg \max_{\mathbf{u}} \rho(\mathbf{u})$.

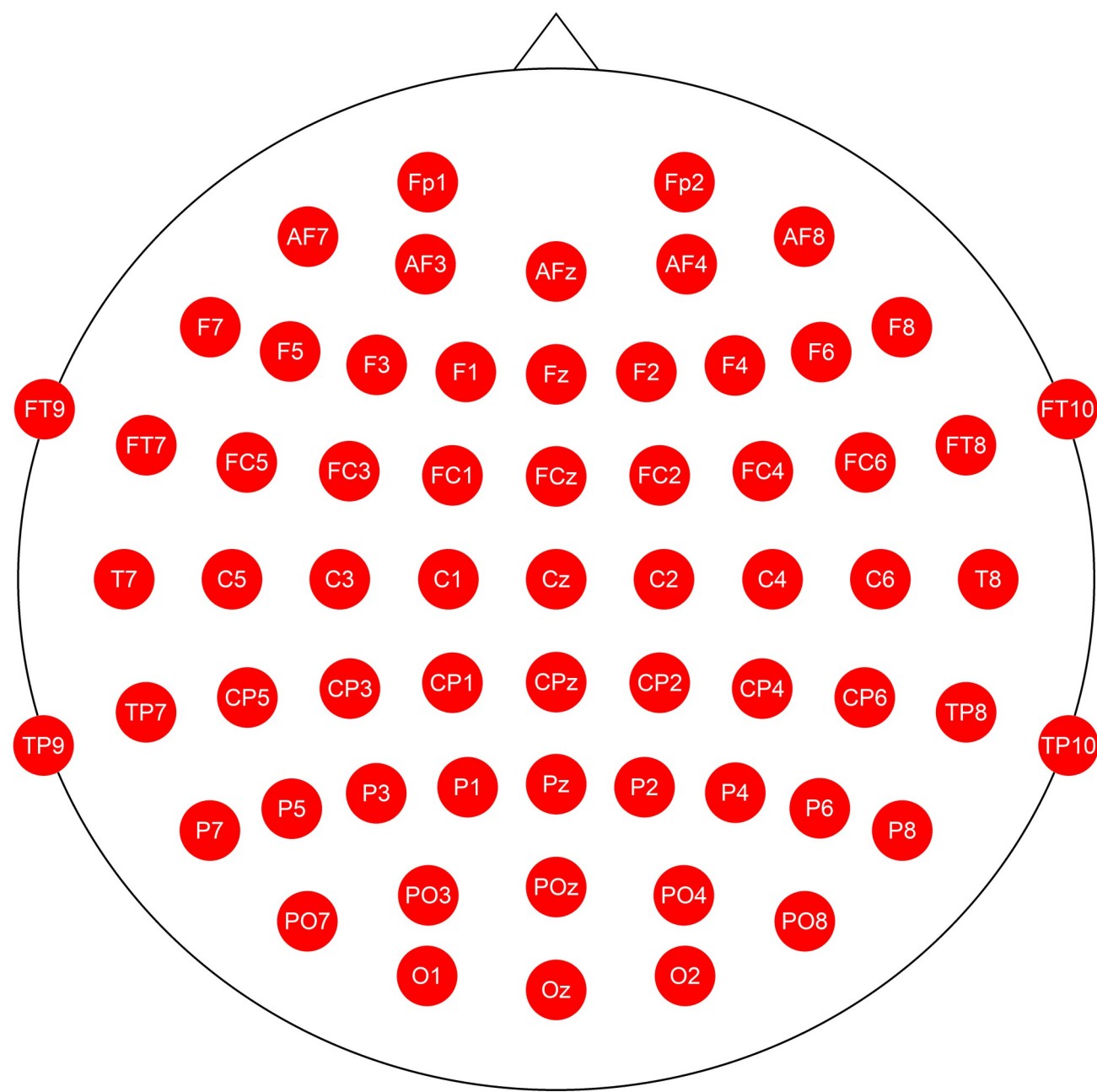

**Fig 3. EEG montage.**

## Classifier based on Riemannian geometry

Barachant et al. [25–27] proposed a novel classifier based on Riemannian geometry. The main idea of this classifier is to identify covariance matrices that have spatial information directly and to extract spatial information without using spatial filtering. By using Riemannian geometry, the distances between covariance metrics can be measured. In this framework, the center of gravity (Riemannian mean) of covariance matrices derived by each epoch in each class is calculated, and the Riemannian distances between the covariance matrix of an unlabeled

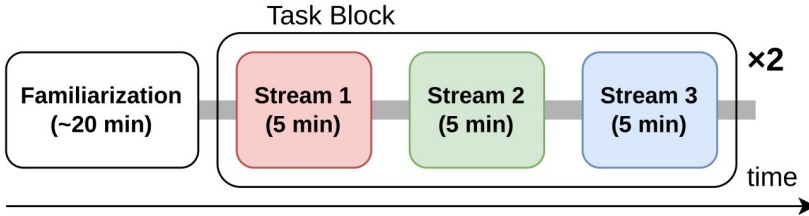

**Fig 4. Time chart of the session.** In a session, firstly, the subject was familiarized with the task. The task block consisted of three runs. In the first, second, and third runs, the subject was requested to attend to Stream 1, 2 and 3, respectively. The task block was conducted two times in total.

epoch and the Riemannian mean of each class are determined. Then, the epoch is labeled as the nearest class. This method is called minimum distance to mean (MDM). To classify covariance matrices by conventional classifiers (e.g., linear discriminant analysis (LDA) and logistic regression), covariance matrices are projected onto the Riemannian tangent space so that they can be manipulated in Euclidean space and vectorized.

Covariance metrics were calculated by the following method [23, 24, 28]. Let $P_1 \in R^{2C \times N_s}$ ($C$ = number of xDAWN components, $N_s$ = number of sample) be the estimated signal subspace derived by the xDAWN filter. Since two xDAWN filters were estimated for both attended and nonattended classes, dimensions of $P_1$ were $2C \times N_s$. Each epoch was defined as $X_i \in R^{2C \times N}$. Then, we calculated a super $\tilde{X}_l \in R^{4C \times N}$ by concatenating $P_1$ and $X_i$.

$$\tilde{X}_l = \begin{bmatrix} P_1 \\ X_i \end{bmatrix} \in R^{4C \times N} \qquad (1)$$

Covariance matrices were built by these supertrials by using the sample covariance matrix (SCM) estimator [26].

$$\tilde{\Sigma}_l = \frac{1}{N-1} \tilde{X}_l \tilde{X}_l^\top \qquad (2)$$

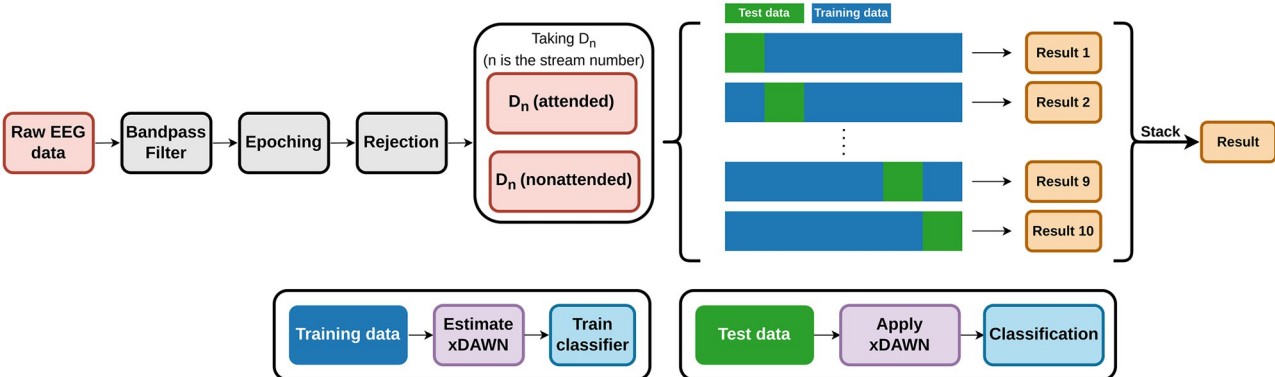

**Fig 5. The flowchart of classification pipeline.** To detect the selective attention to the each stream, the responses to the deviant stimuli $D_n$ in stream $n$ was classified. The classification was done for every $n \in [1, 2, 3]$.

The Riemannian distance between two covariance matrices can be computed by the following equation [25]. Where $\lambda_i$, $i = 1 \ldots 4C$ are the real eigenvalues of $\Sigma_1^{-1}\Sigma_2$.

$$\delta_R(\Sigma_1, \Sigma_2) = \left[\sum_{i=1}^{4C}\log^2\lambda_i\right]^{1/2} \tag{3}$$

Then, the Riemannian mean of the covariance matrices will be computed by the following [25].

$$\Sigma_{\mathfrak{G}} = \mathfrak{G}(\Sigma_1, ..., \Sigma_I) = \arg\min_\Sigma\sum_{i=1}^{I}\delta_R^2(\Sigma, \Sigma_i) \tag{4}$$

Each covariance matrix $\tilde{\Sigma}_i$ can be vectorized by projecting onto Riemannian tangent space to classify with a conventional classifier [25].

$$v_i = \text{upper}\left(\Sigma_{\mathfrak{G}}^{-\frac{1}{2}}\text{Log}_{\Sigma_{\mathfrak{G}}}(\Sigma_i)\Sigma_{\mathfrak{G}}^{-\frac{1}{2}}\right) \tag{5}$$

The feature vector $v_i$ was classified by logistic regression. Three binary classifiers were built to detect whether each stream was attended or not. Each stream consisted of $D_n$ and $S_n$, and all responses to $D_n$ were used for classification where responses to $S_n$ were not. Responses to $D_n$ when selective attention to Stream $n$ was paid and responses when it was not paid were used, and it was classified whether each stream was attended or not (binary classification). The classification performance was evaluated by 10-fold cross validation. MNE-Python (0.23.3) [29], Pyriemann (0.2.7) [30], and scikit-learn (0.23.2) [31] were used to implement the classifier.

The classification results were evaluated by two matrices, accuracy and MCC (Matthews correlation coefficient) [32, 33]. MCC can be derived by following formula, and it takes a value from −1 (worst) to 1 (best). When it takes 0, the output is the random answer.

$$MCC = \frac{TP \times TN - FP \times FN}{\sqrt{(TP + FP)(TP + FN)(TN + FP)(TN + FN)}} \tag{6}$$

Where $TP$, $TN$, $FP$, and $FN$ denote true positives, true negatives, false positives, and false negatives, respectively.

The classification results were also evaluated by plotting the averaged confusion matrices. Since we have three classifiers for detecting the selective attention to each stream, three confusion matrices were derived. For each subject and class, a confusion matrix was derived from the results of 10-fold cross validation, and it was averaged over the subjects.

## Results and discussions

Fig 6 shows averaged responses (electrode Cz) to target stimuli when subject A attended Stream 1 (red), Stream 2 (green), and Stream 3 (blue). Fig 6(a)–6(c) shows responses to target stimuli corresponding to Streams 1, 2, and 3, respectively. The gray boxes denote the significant difference between responses when subjects attended the corresponding stream and responses when subjects attended a noncorresponding stream. For subject A, P300 activity was elicited by target stimuli with a latency of approximately 350 ms when attending to Stream 1 and a latency of approximately 300 ms when Stream 2 or 3 was attended. The peak amplitudes were quite large, and P300 activity was elicited only by the target stimuli in the attended stream.

For subject B, P300 activity was elicited by target stimuli with a latency of approximately 250 ms when attending to Stream 2 or 3. MMN was elicited by the target stimuli in the

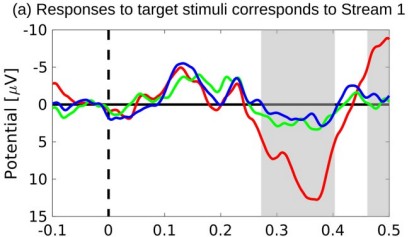
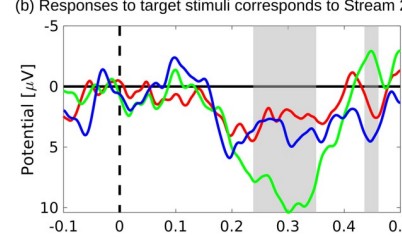
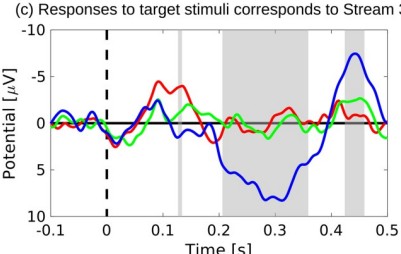

**Fig 6. Averaged responses (electrode Cz) to target stimuli when subject A attended to each stream.** In each figure, red, green, and blue denote Stream 1, Stream 2, and Stream 3. Responses to target stimuli corresponding to Stream 1, 2, and 3 are shown in (a), (b), and (c), respectively. Gray boxes show the t-test results and denote the significant difference between responses when subjects attended to the corresponding stream and did not attend to the corresponding stream. (Student's t-test $p < 0.01$).

attended stream when the subject attended to Stream 1. For subject C, positive responses were observed in the occipital region with a latency of 250–500 ms. The activity cannot be concluded as P300, but it was elicited only by the target stimuli corresponding to the attended stream. For subject G, P300 activity with small amplitudes was elicited by the target stimulus when attending to Stream 1; however, no significant response was observed when the subject attended to Streams 2 and 3.

Fig 7 shows averaged responses that were averaged over all eight subjects. Red, blue, and green lines denote responses to target stimuli when subjects attended Streams 1, 2, and 3, respectively. Scalp topographies of each latency are also shown. Fig 7(a)–7(c) show responses to target stimuli corresponding to Streams 1, 2, and 3, respectively. The gray boxes show the t-test results and denote the significant difference between responses when subjects attended the corresponding stream and responses when subjects attended a noncorresponding stream.

As shown in Fig 7(a), positive responses with the latency of approximately 300 400 ms had parietally maximal scalp topographies of P300 activity. Responses when subjects attended Stream 2 had smaller amplitudes. Responses when subjects attended Stream 3 also had smaller amplitudes and frontally maximal scalp topographies. At a latency of 100–250 ms, the amplitude of MMN responses when the subject attended to Stream 1 was significantly larger than when the subject attended to Stream 2 or 3. According to Fig 7(b), P300 activity could be observed clearly at a latency of approximately 250 ms. According to the scalp topographies, responses when the subject attended to Stream 2 had a maximum parietal amplitude, and responses when the subject attended to Stream 1 or 3 had a larger frontal amplitude. According to Fig 7(c), P300 responses with a latency of approximately 250 ms when subjects attended to Stream 3 were significantly larger than responses when subjects attended to Stream 1 or 2. According to the scalp topographies, although responses when subjects attended to Stream 3 had maximum amplitude on the parietal side, responses when subjects attended to Stream 1 or 2 had maximum amplitude on the frontal side. All P300 responses to target stimuli tended to have parietally maximal scalp topographies when subjects attended to the corresponding stream and had frontal maximal scalp topographies when subjects attended to a noncorresponding stream.

We considered that parietally maximal responses were P3b, which reflects the subject's selective attention, and frontally maximal responses were P3a, which is evoked exogenously and does not reflect the subject's selective attention [5].

In five out of eleven subjects, P300 responses were elicited by the target stimulus only when subjects attended to the corresponding stream. Furthermore, in five subjects, the amplitude of

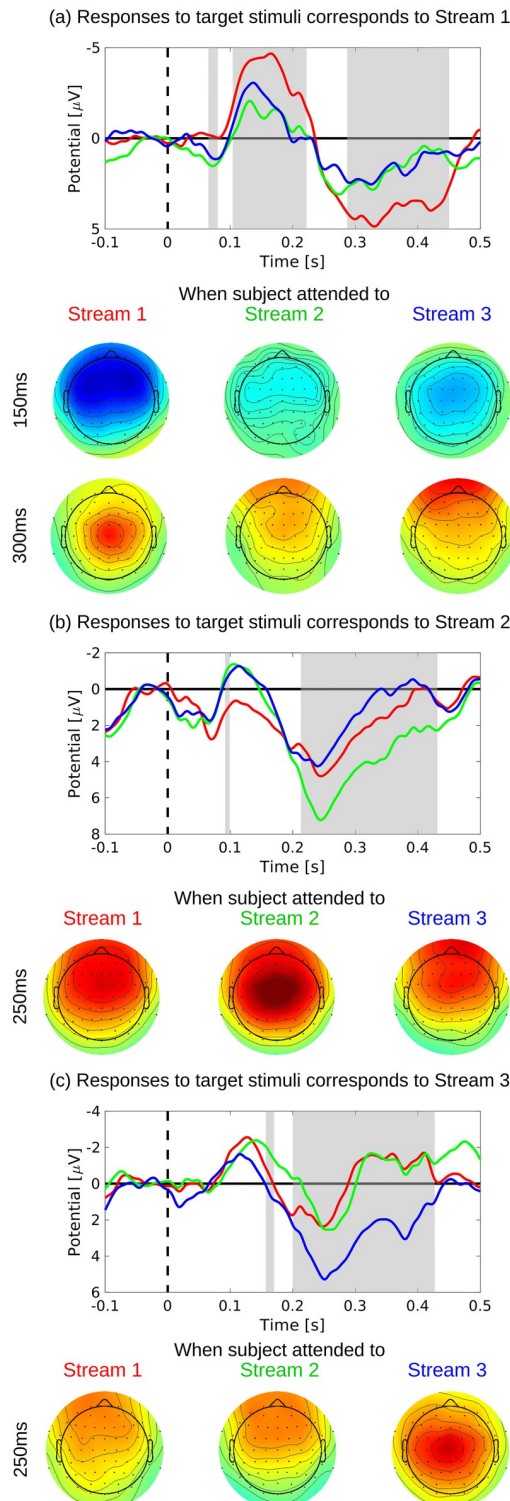

**Fig 7. Averaged responses averaged over all eleven subjects.** Red, blue, and green lines denote responses to target stimuli when subjects attended to Stream 1, 2, and 3, respectively. Scalp topographies of each latency are also shown. Responses to target stimuli corresponding to Stream 1, 2, and 3 are shown in (a), (b), and (c), respectively. Gray boxes show the t-test results and denote the significant difference between responses when subjects attended to the corresponding stream and when they attended to a noncorresponding stream.

**Table 2. Classification results.**

| Subject | Stream 1 | | Stream 2 | | Stream 3 | | Average | |
|---|---|---|---|---|---|---|---|---|
| | Accuracy | MCC | Accuracy | MCC | Accuracy | MCC | Accuracy | MCC |
| A | 0.86 | 0.69 | 0.76 | 0.43 | 0.82 | 0.59 | 0.82 | 0.57 |
| B | 0.91 | 0.80 | 0.83 | 0.62 | 0.91 | 0.75 | 0.88 | 0.73 |
| C | 0.78 | 0.51 | 0.76 | 0.42 | 0.82 | 0.58 | 0.79 | 0.50 |
| D | 0.80 | 0.51 | 0.78 | 0.48 | 0.80 | 0.46 | 0.79 | 0.48 |
| E | 0.81 | 0.55 | 0.76 | 0.46 | 0.80 | 0.54 | 0.79 | 0.52 |
| F | 0.79 | 0.48 | 0.79 | 0.43 | 0.81 | 0.35 | 0.80 | 0.42 |
| G | 0.79 | 0.51 | 0.72 | 0.34 | 0.71 | 0.30 | 0.74 | 0.39 |
| H | 0.78 | 0.51 | 0.74 | 0.39 | 0.73 | 0.36 | 0.75 | 0.42 |
| I | 0.87 | 0.70 | 0.89 | 0.74 | 0.80 | 0.55 | 0.85 | 0.66 |
| J | 0.70 | 0.33 | 0.61 | 0.09 | 0.70 | 0.26 | 0.67 | 0.23 |
| K | 0.90 | 0.76 | 0.83 | 0.54 | 0.73 | 0.39 | 0.82 | 0.56 |
| Grand Average | 0.82 | 0.58 | 0.77 | 0.45 | 0.78 | 0.47 | 0.79 | 0.50 |

MMN responses elicited by the target stimulus was larger when the subject attended to the corresponding stream than when the subject attended to the noncorresponding streams.

Table 2 shows the classification result of each subject, and Fig 8 shows the averaged classification scores for each subject and each stream. The averaged classification accuracy was over 80% for five subjects, 75% − 79% for four subjects and 65% − 74% for two subject. In subject G, as mentioned above, no significant response was observed unless Stream 1 was attended, and the average accuracy was 74%. The classification score was the lowest for subject J. The classification accuracy was lower in some subjects than in others. From ERP plos for these subjects, it was found that P300 was elicited for the deviant stimuli which corresponds to nonattended stream, or the amplitude of P300 elited by deviant stimuli corresponds to attended stream was quite small. The possible reason for these results may be that subjects could not perceive the presented sequences as three segregated streams. In the absence of stream perception, it may be difficult to find the deviant stimuli corresponds to the target stream, or the the subjects may attend not only to deviant stimuli corresponds to target stream but to deviant stimuli corresponding to all streams. In subject B, average accuracy reached 88%. When this subject attended Stream 1, only MMN was elicited by the target stimuli, and P300 was not. However, 91% accuracy was obtained when Stream 1 was attended. S1 Fig Shows a grand averaged confusion matrix. In this study, three classifiers were trained, and the confusion matrix for each is shown. All confusion matrices from eleven subjects were averaged over subjects. The accuracy averaged over all subjects when Streams 1, 2, and 3 were attended were 82%, 77%, and 78%, respectively. According to this result, the accuracy for Stream 2 was slightly low. A few subjects mentioned that attending Stream 2 was harder than attending the other streams, and this is consistent with the result.

The average accuracy of overall results was 79%. In the proposed system, since the required time to present target stimuli in all streams was inconsistent, the classification interval fluctuated in the range of 0.54 − 9.18 s with a certain probability, and its expected value was 7.42 s. Assuming that 3-class classification was performed with an accuracy of 79% every 7.42 s, the information transfer rate (ITR) could be calculated as 5.12 bits/min. The ITR will be in the range of 4.14 − 70.39 bits/min depending on the classification interval when accuracy is consistent at 79%. Due to fluctuations in the classification interval, it is not appropriate to compare this system with other systems; however, our previous research achieved an ITR of

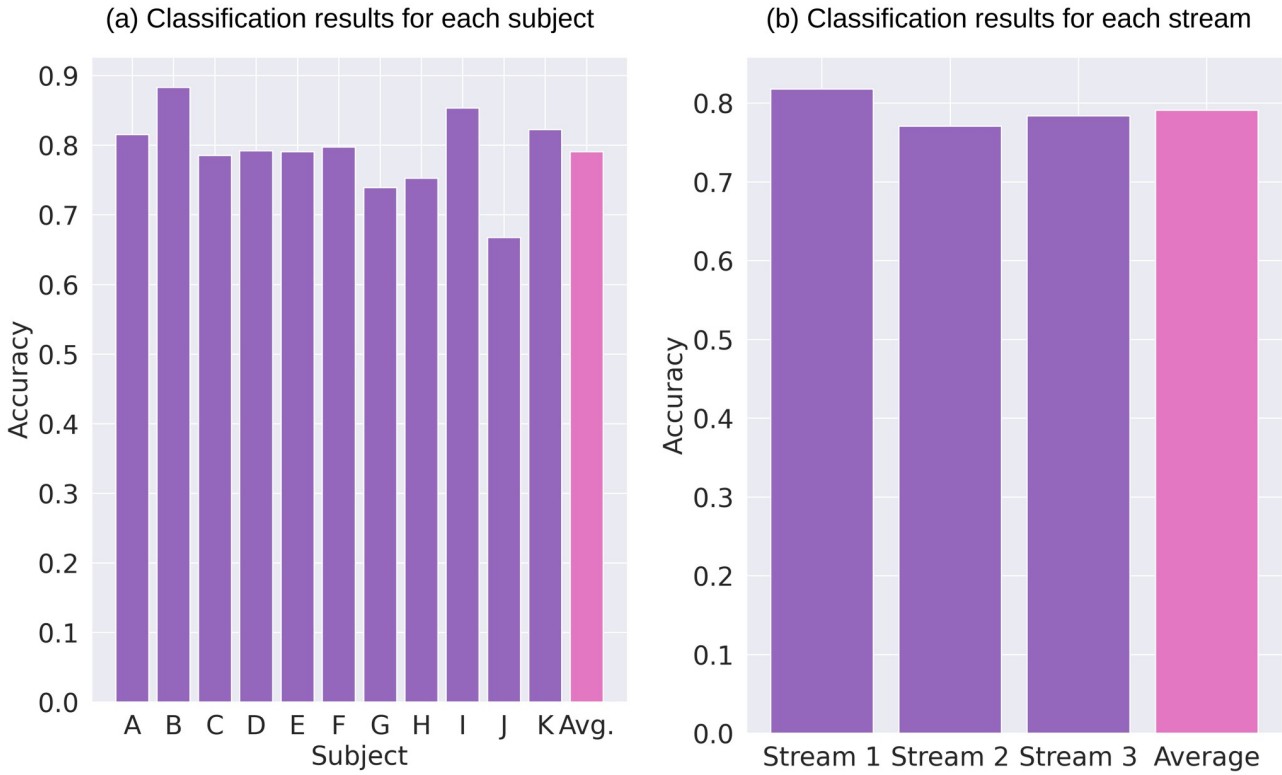

**Fig 8. Classification results.** (a) Averaged classification accuracy for each subject. (b) Averaged classification accuracy for each stream.

approximately 5 bits/min (in this system, the classification interval was fixed to 10 s), and in the system developed by Schreuder et al. [12], average ITR of 17.39 bits/min was achieved. In the proposed system, minimizing the classification interval (e.g., by using shorter SOAs or modifying the probabilities of target and nontarget stimuli) is effective to improve the ITR; however, these approaches may also cause undesired modulations of ERP responses. Hence, optimization is the key to improving ITR and will be studied in future work.

In this study, one female out of 11 subjects participated, and the gender distribution of subjects was uneven. Only limited research has been done on evaluating the influence of gender on BCI performance; it was reported that the performance of the motor-imagery-based and visual P300-based BCI is higher in females [34] and males [35], respectively. There is no such report on auditory BCI, and some other auditory BCI researches have an uneven gender distribution [12, 15, 36, 37]. Regarding auditory P300, it was reported that the amplitude of P300 tends to be higher in females, and the P300 latency is comparable between genders [38]. On auditory scene perception, some studies showed the better performance of males than females for localizing target sounds in a multi-source sound environment [39, 40]; however, they also pointed out that there was a large interindividual variability [40], and there is no clear conclusion. The effects of gender on auditory stream segregation are not yet known. The promising results were shown with a number of subjects in this study. Still, further research is required to investigate the influence of gender differences in the performance of BCI based on auditory stream segregation.

The proposed system can be used by presenting stimuli to a person's single ear, and patients who are deaf in one ear can use the system. Furthermore, this system does not require many

speakers and a multichannel audio interface but only requires headphones and an audio interface with more than one channel. This makes the system easier to use and less expensive; moreover, it has potential for practical and medical usage. In this paper, 64-channel EEG signals of participants were recorded; however, this number of signals is too large for practical usage. Since the xDAWN filter was applied, the number of channels can be reduced based on the contribution of each channel in the spatial filter; moreover, and the channel reduction method based on the xDAWN algorithm, which was proposed by Rivet et al. [24], can be a possible option. Furthermore, a sophisticated machine learning method can be used to improve ITR and robustness. Additionally, optimizing the parameters of the tone itself and its sequences (e.g., frequency, tone, and SOA) can enhance users' ability to discriminate among streams and can improve the system.

## Conclusion

In this study, three oddball sequences consisting of musical tones were presented to each subject's right ear. Subjects were asked to pay attention to one of the presented sequences and count the number of target stimuli, and responses to each target stimulus were analyzed and classified based on Riemannian geometry. P300 activity was elicited by the subject's selective attention to the tone stream, and the subject's attended stream could be detected with high accuracy by classifying the responses elicited by the target stimuli in each stream.

Multiclass auditory systems that have been proposed to date mainly use the location of sound sources to make a variety of auditory stimuli [11, 12]. Hence, these systems do not make the best use of properties of tones, such as frequency, intensity, or timbre. In our previous research, an auditory illusion called stream segregation was tested, and its result was promising; however, it had allowed for a binary decision to be made. In this study, by utilizing musical tones, the BCI system based on auditory stream segregation was extended to three classes. The present results indicate that auditory stimuli based on stream segregation can be used by a multiclass auditory BCI system and enhance current systems.

## Supporting information

**S1 Fig. Grand averaged confusion matrix.** From the classification results from 10-fold cross-validation for each subject, a confusion matrix was derived, and the confusion matrices from all subjects were averaged over subjects.
(TIF)

## Author Contributions

**Conceptualization:** Simon Kojima, Shin'ichiro Kanoh.

**Data curation:** Simon Kojima.

**Formal analysis:** Simon Kojima.

**Funding acquisition:** Shin'ichiro Kanoh.

**Investigation:** Simon Kojima.

**Methodology:** Simon Kojima, Shin'ichiro Kanoh.

**Project administration:** Shin'ichiro Kanoh.

**Resources:** Shin'ichiro Kanoh.

**Software:** Simon Kojima.

**Supervision:** Shin'ichiro Kanoh.

**Validation:** Simon Kojima, Shin'ichiro Kanoh.

**Visualization:** Simon Kojima.

**Writing – original draft:** Simon Kojima.

**Writing – review & editing:** Shin'ichiro Kanoh.

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
