## [Decision Letter · Decision Letter 0]

12 Nov 2023

PONE-D-23-26472An auditory brain-computer interface based on selective attention to multiple tone streamsPLOS ONE

Dear Dr. Kojima,

Thank you for submitting your manuscript to PLOS ONE. After careful consideration, we feel that it has merit but does not fully meet PLOS ONE’s publication criteria as it currently stands. Therefore, we invite you to submit a revised version of the manuscript that addresses the points raised during the review process.

We look forward to receiving your revised manuscript.

Kind regards,

Anwar P.P. Abdul Majeed

Academic Editor

PLOS ONE

Journal Requirements:

   "This work was supported by JSPS KAKENHI Grant Number JP23K11811"

   "Initials: S. Kanoh

Grant number : JP23K11811

Funder: JSPS KAKENHI

URL: https://www.jsps.go.jp/

Reviewers' comments:

Reviewer's Responses to Questions

**Comments to the Author**

1. Is the manuscript technically sound, and do the data support the conclusions?

Reviewer #1: Yes

Reviewer #2: Partly

2. Has the statistical analysis been performed appropriately and rigorously? 

Reviewer #1: N/A

Reviewer #2: No

3. Have the authors made all data underlying the findings in their manuscript fully available?

Reviewer #1: No

Reviewer #2: Yes

4. Is the manuscript presented in an intelligible fashion and written in standard English?

Reviewer #1: Yes

Reviewer #2: Yes

5. Review Comments to the Author

Reviewer #1: The authors proposed a three-class BCI using three auditory tones, and tested on a small number of healthy subjects. The manuscript is well written and technical sound. However, there are still a few concerns:

1. the number of subjects is small and without any testing on patients. It is therefore strongly recommended to recruit some more subject, if not patients, given the relatively simple paradigm.

2. The authors mentioned auditory BCIs for ALS, but missed some key publications in this field, such as 10.3389/fnins.2017.00251, and 10.1007/978-981-16-5540-1_34 .

3. did any of the subjects have experience with auditory BCI? and if there was a testing session where the subjects can learn the paradigm?

4. the image quality needs to be improved, as they look blurred in the pdf

5. it is recommended to report the accuracy from each stream, so as to show if there is any imbalance among streams.

Reviewer #2: Overall, the contents of the manuscript just fair or average. Author needs to consider the following comments in order to enhance the quality of the manuscript.

Abstract section:

i) Author needs to mention the selected EEG features to achieve the classification accuracy > 80% for 3 subjects and > 75% for 4 subjects. Author needs to explain why the accuracy drop to 75% for 4 subjects.

Introduction section:

i) Author had mentioned in this section , "The largest amplitude is observed on the vertex" (page 2/9). Author needs to explain this statement. What is the value of the amplitude?

ii) The literature review on the Auditory BCI is good. However, there is only 10 citations from the references in this section which is not enough for the standard of the journal. It is suggested to author to cite more latest references such as in year 2020 till 2023.

Materials and Method section:

i) It is suggested to author to start this section with the process flowchart of the study.

ii) Author needs to create and implement the confusion matrix for the classification process.

iii) Author needs to explain how the selection of the EEG features are done.

iv) It is much preferable if the author can provide the sketch of the EEG montage to show the location of the EEG measurement.

v) It is good that author had provided the evidence of the work ethic approval.

vi) The procedure of the statistical analysis is not clearly described.

vii) Author needs to share the example of the xDAWN filter and explain the procedure of using this filter.

viii) It is only 2 equations are provided which is for calculating Sample Covariance Matrix (SCM). Author needs to provide formula equation for other processes such as 10-fold cross validation and t-test, and EEG signals features ERP, SSNR, Riemannian geometry and MCC.

ix) Author needs to verify this statement, "Recorded signals were bandpass filtered at 1 Hz to 40 Hz (FIR filter) at page 4/9. It is contradicted with the previous paragraph where author have mentioned that "Recorded signals were bandpass filtered at 1 Hz to 40 Hz (2nd Butterworth filter). It is because Butterworth filter IS NOT FIR filter.

x) Author needs to defined MCC and share formula of MCC.

Results and Discussion section:

i) The results are presented in term of the response to target stimuli as shown in Figure 2 and Figure 3 and also

classification results in Table 2. The explanation of the figures and table are good. How about statistical results? How about the 10-fold cross validation results? How about the statistical analysis results, t-test?

ii) What is the results of the confusion matrix for the classification process?

iii) It is suggested to author to represent the results in term of graph for better view and understanding.

iv) Author needs to explain the results of the >80% and >75% classification accuracy.

Acknowledgement section:

Good.

References section:

i) 18 references are not good enough since the manuscript will be sent to Journal. It is suggested to author to cite more than 30 papers including the latest paper, such as published in year 2020 till 2023.

ii) There are very old references which are reference # 2 (published in year 1988), reference # 9 (published in year 1990).

6. PLOS authors have the option to publish the peer review history of their article (what does this mean?). If published, this will include your full peer review and any attached files.

Reviewer #1: No

Reviewer #2: No

---

## [Author Response · Author response to Decision Letter 0]

27 Dec 2023

Thank you for inviting us to submit a revised manuscript. We have noted all responses to each feedback and comment in the submitted "Response to Reviewers" file. We would like to ask you to confirm it. We hope that our edits and the responses we provide satisfactorily address all the issues and concerns you and the reviewers have noted.

---

## [Decision Letter · Decision Letter 1]

5 Feb 2024

PONE-D-23-26472R1An auditory brain-computer interface based on selective attention to multiple tone streamsPLOS ONE

Dear Dr. Kojima,

Thank you for submitting your manuscript to PLOS ONE. After careful consideration, we feel that it has merit but does not fully meet PLOS ONE’s publication criteria as it currently stands. Therefore, we invite you to submit a revised version of the manuscript that addresses the points raised during the review process.

We look forward to receiving your revised manuscript.

Kind regards,

Anwar P.P. Abdul Majeed

Academic Editor

PLOS ONE

Journal Requirements:

Reviewers' comments:

Reviewer's Responses to Questions

**Comments to the Author**

1. If the authors have adequately addressed your comments raised in a previous round of review and you feel that this manuscript is now acceptable for publication, you may indicate that here to bypass the “Comments to the Author” section, enter your conflict of interest statement in the “Confidential to Editor” section, and submit your "Accept" recommendation.

Reviewer #1: All comments have been addressed

Reviewer #2: (No Response)

2. Is the manuscript technically sound, and do the data support the conclusions?

Reviewer #1: Yes

Reviewer #2: Yes

3. Has the statistical analysis been performed appropriately and rigorously? 

Reviewer #1: Yes

Reviewer #2: Yes

4. Have the authors made all data underlying the findings in their manuscript fully available?

Reviewer #1: Yes

Reviewer #2: Yes

5. Is the manuscript presented in an intelligible fashion and written in standard English?

Reviewer #1: Yes

Reviewer #2: Yes

6. Review Comments to the Author

Reviewer #1: The authors address my comments properly, and to my point of view, the manuscript is ready for publishing.

Reviewer #2: Overall content of the revised manuscript looks much better than before. Author had just addressed some of the comments from the first review of the manuscript.

Abstract section:

i)The number of subject is increased to 11. Why only 1 female participant in the study?

ii) Author had explained the difference in the classification accuracy for 5 subjects and 9 subjects which it is acceptable.

Introduction section:

i)The content of this section in the revised manuscript look much better than before. However, it seems that this section is quite short. It is suggested to author to describe more on ERP by citing more references related to ERP.

ii) The literature reviews on Auditory BCIs are good which citation comes from recent publication.

Materials and Method section:

i) Author just add time chart of the experimental work which just describe the duration of each task. It is not enough. Thus, it is highly recommended to author to include flowchart of the study as well.

ii)The confusion matrices are still not provided in the manuscript.

iii) The process of generating feature are clearly explained.

iv) The diagram of EEG montage is provided as shown in Fig 2.

v) Author had changed the filter type from FIR filter to 2nd order Butterworth filter and filter range from 0.1 Hz to 40 Hz. However, author needs to explain why the 2nd order Butterworth is employed to filter the EEG and EOG data.

vi) The use of statistical analysis (t-test) is described in this section.

vii) The detail description of the xDAWN filter is provided in this section.

viii) The explanation on Reimannian classifier, 10-fold cross validation, MCC and t-test are provided. However, author still need to explain the variables in the formula such as TP, TN, FP & FN.

Results and Discussions section:

i) The statistical results of ERP (t-test) are provided in this section.

ii) Table 2 show the classification accuracy and MCC NOT the 10-fold cross validation results.

iii) The results of confusion matrix is provided in S1 Fig.

iv) The classification results for each subject and stream are provided.

v) The accuracy drop for some subjects are also explained.

Acknowledgement section:

Good

References section:

i) The number of references are good (32 references which include the recent publication).

ii) The explanation for selecting old references are acceptable.

7. PLOS authors have the option to publish the peer review history of their article (what does this mean?). If published, this will include your full peer review and any attached files.

Reviewer #1: **Yes: **Ren Xu

Reviewer #2: No

---

## [Author Response · Author response to Decision Letter 1]

6 Feb 2024

Responses to Reviewer #1

1. The authors address my comments properly, and to my point of view, the manuscript is ready for publishing.

 We appreciate your insightful and constructive feedback on our manuscript. We believe that this revision has further improved the quality of the paper.

Responses to Reviewer #2

We appreciate your insightful and constructive feedback on our manuscript. We incorporated all comments and believe the quality of the paper has improved a lot. The following are responses from the items for which responses were requested.

# Abstract section:

1. The number of subject is increased to 11. Why only 1 female participant in the study?

 Among the participants, we measured the data from one female, and it was by chance. There were no reports of gender differences in acoustic scene perception that we could find. In addition, we believe that factors such as musical experience are more important in this experiment and that gender is not the main factor influencing the experiment results. 

# Introduction section:

1. The content of this section in the revised manuscript look much better than before. However, it seems that this section is quite short. It is suggested to author to describe more on ERP by citing more references related to ERP.

 We added citations regarding ERP and described P300 ERP in detail in the Introduction section. (lines 10-24)

# Materials and Method section:

1. Author just add time chart of the experimental work which just describe the duration of each task. It is not enough. Thus, it is highly recommended to author to include flowchart of the study as well.

 We added the two figures, Fig 1 and 5, to explain the study. Fig 1 shows the conceptual diagram of the study, and Fig 5 shows the classification pipeline.

2. The confusion matrices are still not provided in the manuscript.

 We added the description of the confusion matrices in the Materials and Method section.(lines 212-214)

3. Author had changed the filter type from FIR filter to 2nd order Butterworth filter and filter range from 0.1 Hz to 40 Hz. However, author needs to explain why the 2nd order Butterworth is employed to filter the EEG and EOG data.

 We received a comment from you in the previous review that it contradicted the description of the filter. In the first manuscript, we used the FIR filter for the classification pipeline and the Butterworth filter for plotting ERPs. This was because of the implementation. We generally use the Butterworth filter. However, MNE-python, which is used to implement the classification pipeline, uses the FIR filter by default. However, it is suggested to use IIR filters, including Butterworth, since it does not need to use as many time points as FIR (Luck et al., 2014). Additionally, it is known that Butterworth has a more linear phase response compared to other filters. For these reasons, we changed the filtering method of the classification pipeline from FIR to Butterworth. 

Regarding the filter range, (Luck et al., 2014) mentioned that “applying a high-pass filter with 0.1Hz provides the best balance between statistical power and waveform distortion for the P3 wave”. He also suggested applying a low-pass filter with 30Hz. However, we can see the shape of the responses in detail with a higher cutoff frequency of a low-pass filter. In the area where our institution is located, the power line frequency is 50Hz. Thus, we decided to use 40Hz for low-pass filtering.

We have explored some articles related to brain-computer interfaces published on PLOS ONE and found that these are not discussing regarding the filter type and range in details. Hence we have decided we don’t include the description above.

4. The explanation on Reimannian classifier, 10-fold cross validation, MCC and t-test are provided. However, author still need to explain the variables in the formula such as TP, TN, FP & FN.

 We added descriptions about TP, TN, FP, and FN. (lines 210-211)

# Results and Discussion section:

1. Table 2 show the classification accuracy and MCC NOT the 10-fold cross validation results.

 The classification accuracy and MCC shown in Table 2 were obtained by a 10-fold cross-validation procedure. We derived classifier output as predicted labels with 10-fold cross-validation. The accuracy and MCC were obtained with the predicted labels and true labels.

---

## [Decision Letter · Decision Letter 2]

27 Mar 2024

PONE-D-23-26472R2An auditory brain-computer interface based on selective attention to multiple tone streamsPLOS ONE

Dear Dr. Kojima,

Thank you for submitting your manuscript to PLOS ONE. After careful consideration, we feel that it has merit but does not fully meet PLOS ONE’s publication criteria as it currently stands. Therefore, we invite you to submit a revised version of the manuscript that addresses the points raised during the review process.

We look forward to receiving your revised manuscript.

Kind regards,

Anwar P.P. Abdul Majeed

Academic Editor

PLOS ONE

Journal Requirements:

**Additional Editor Comments:**

I have reviewed the correspondance between Reviewer 2 and the authors, and the following is my advise to wrap this up:

1. Perhaps it is best that you could cite other works that has uneven number of gender distribution from the participants or perhaps omit the female participant from the study and in the discussion/conclusion/future works section, suggest that future studies would be look into the influence of gender difference towards the results.

2. For figure 5, you may retain it as is (don't need to change to a process flow-chart)

3. Find a reference or references to support the use of the second order filter

4. I am happy with the 10-fold cross validation results that have been presented (but you may need to change it if you were to use 10 participants).

Reviewers' comments:

Reviewer's Responses to Questions

**Comments to the Author**

1. If the authors have adequately addressed your comments raised in a previous round of review and you feel that this manuscript is now acceptable for publication, you may indicate that here to bypass the “Comments to the Author” section, enter your conflict of interest statement in the “Confidential to Editor” section, and submit your "Accept" recommendation.

Reviewer #2: (No Response)

2. Is the manuscript technically sound, and do the data support the conclusions?

Reviewer #2: Yes

3. Has the statistical analysis been performed appropriately and rigorously? 

Reviewer #2: Yes

4. Have the authors made all data underlying the findings in their manuscript fully available?

Reviewer #2: Yes

5. Is the manuscript presented in an intelligible fashion and written in standard English?

Reviewer #2: Yes

6. Review Comments to the Author

Reviewer #2: Overall contents of the manuscript looks much better compared to the previous version of the manuscript. However, some of comments still not addressed properly by the authors as stated below;

Abstract section:

I am not agree with your statement that gender is not the main factor in your study in acoustic scene perception and experimental results. I suggest you to provide variation in your experimental works by including more female subjects. You also cannot said that the participation of the female subject is by "chance".

Introduction section:

It is good that you have addressed my comments by describing more on P300 ERP and add citations for ERP.

Materials and Method section:

1.Authors have not addressed my comments properly.

Fig. 1 can be considered as block diagram of the study which is good. However, Fig. 5 also looks like block diagram as well. I am much prefer if you can provide the process flowchart of the study (top-down process) which shows the detail processes and parameters that are involved in the study.

2. The operations of the confusion matrices are not well explained in the manuscript as stated in line 212-214.

3. You need to write, "... (2nd order Butterworth IIR filter)" in line 154 of the manuscript. However, authors still die not explain the selection of 2nd order Butterworth IIR filter. Why 2nd order? Author needs to provide more precise statement in choosing 2nd order filter.

4. It is good that the descriptions of TP, TN, FP and FN are provided at line 210-211.

Results and Discussion section:

Authors have still not addressed the comments of this section precisely. Authors have written that "The classification performance was evaluated by 10-fold cross validation." How author implement the 10-fold cross validation, there is no formula provided to perform this technique.

For my knowledge, classification accuracy and 10-fold cross validation are different thing. Classification accuracy is obtained by applying several classifiers such as SVM, kNN, ANN and others. However, in your study, you have applied Riemannian geometry as shown in Table 2. However, the 10-fold cross validation is one technique to validate the accuracy obtained using Riemannian geometry classifier.

7. PLOS authors have the option to publish the peer review history of their article (what does this mean?). If published, this will include your full peer review and any attached files.

Reviewer #2: No

---

## [Author Response · Author response to Decision Letter 2]

5 Apr 2024

# Responses to Editor

We appreciate your insightful and constructive suggestions and comments on our manuscript. We incorporated all comments and believe the quality of the paper has improved a lot.

 1. Perhaps it is best that you could cite other works that has uneven number of gender distribution from the participants or perhaps omit the female participant from the study and in the discussion/conclusion/future works section, suggest that future studies would be look into the influence of gender difference towards the results.

Regarding auditory BCI, it has not been reported that gender differences influence its performance. As stated in the manuscript, some famous studies published in journals did not balance the gender distribution of participants (lines 313 - 314), and we consider it does not necessarily have to be balanced. 

It was reported that the amplitude of auditory P300 tends to be larger in females. If this is true, the performance would be higher, making the gender distribution even since we did the experiment on more males. However, there is no clear conclusion on it. It is required to investigate the influence of gender differences on the performance of auditory BCI based on stream segregation, and we have mentioned it in the manuscript. Lines(309 - 322)

2. For figure 5, you may retain it as is (don't need to change to a process flow-chart)

We decided to retain it as it is.

3. Find a reference or references to support the use of the second order filter

Luck et al. (2014, “An Introduction to the Event-Related Potential Technique 2nd edition”) recommended to use a filter with a slope between 12 and 24 dB/octave for offline ERP analysis. The 2nd order Butterworth filter has a slope 12 dB/octave. To achieve zero-phase filter, we applied this filter twice, once forward and once backwards, and it made a slope 24 dB/octave. We cited the book and added the description why we selected 2nd order filter. (lines 135 - 139)

4. I am happy with the 10-fold cross validation results that have been presented (but you may need to change it if you were to use 10 participants).

We used 10-fold cross validation to evaluate accuracy for each subject, and it was not used for subject to subject transfer learning. 

# Responses to Reviewer 2

We appreciate your insightful and constructive feedback on our manuscript. We incorporated all comments and believe the quality of the paper has improved a lot. The following are responses from the items for which responses were requested.

Abstract section:

1. I am not agree with your statement that gender is not the main factor in your study in acoustic scene perception and experimental results. I suggest you to provide variation in your experimental works by including more female subjects. You also cannot said that the participation of the female subject is by "chance".

Thank you for your comment. Considering the advise from the editor, we have added descriptions of gender differences in ERP, BCI, and auditory perception in the text, and we have stated that gender differences in auditory stream segregation and the performance of the BCI based on it have not been studied and will be investigated as a future work. (lines 309 - 322)

Materials and Method section:

1. Fig. 1 can be considered as block diagram of the study which is good. However, Fig. 5 also looks like block diagram as well. I am much prefer if you can provide the process flowchart of the study (top-down process) which shows the detail processes and parameters that are involved in the study.

Thank you for your comment. We considered the advise from the editor and decided to keep the figure as it is.

2. The operations of the confusion matrices are not well explained in the manuscript as stated in line 212-214.

We added the description how it was obtained in the Materials and Method section. (lines 217 - 219)

3. You need to write, "... (2nd order Butterworth IIR filter)" in line 154 of the manuscript. However, authors still die not explain the selection of 2nd order Butterworth IIR filter. Why 2nd order? Author needs to provide more precise statement in choosing 2nd order filter.

We wrote “(zero-phase 2nd order Butterworth IIR filter, slope 24 dB/octave)”. (line 135 and 157)

 Luck et al. (2014, “An Introduction to the Event-Related Potential Technique 2nd edition”) recommended to use a filter with a slope between 12 and 24 dB/octave for offline ERP analysis. The 2nd order Butterworth filter has a slope 12 dB/octave. To achieve zero-phase filter, we applied this filter twice, once forward and once backwards, and it made a slope 24 dB/octave. We added the description why we selected 2nd order filter. (lines 135 - 139)

Results and Discussion section:

1. Authors have still not addressed the comments of this section precisely. Authors have written that "The classification performance was evaluated by 10-fold cross validation." How author implement the 10-fold cross validation, there is no formula provided to perform this technique. For my knowledge, classification accuracy and 10-fold cross validation are different thing. Classification accuracy is obtained by applying several classifiers such as SVM, kNN, ANN and others. However, in your study, you have applied Riemannian geometry as shown in Table 2. However, the 10-fold cross validation is one technique to validate the accuracy obtained using Riemannian geometry classifier.

In 10-fold cross validation, whole data was divided into 10 blocks, and 9 blocks were used for learning the model, and one block were used as the testing data, and the class label was estimated for each sample in testing data. It was repeated 10 times with changing a block used for learning the model. From the true labels and the predicted labels obtained by 10-fold cross validation, accuracy was derived. We believe this procedure is commonly used in the BCI research field and it’s called “cross validation”.

---

## [Editor Report · Decision Letter 3]

29 Apr 2024

An auditory brain-computer interface based on selective attention to multiple tone streams

PONE-D-23-26472R3

Dear Dr. Kojima,

We’re pleased to inform you that your manuscript has been judged scientifically suitable for publication and will be formally accepted for publication once it meets all outstanding technical requirements.

Kind regards,

Anwar P.P. Abdul Majeed

Academic Editor

PLOS ONE

---

## [Editor Report · Acceptance letter]

9 May 2024

PONE-D-23-26472R3 

PLOS ONE

Dear Dr. Kojima, 

I'm pleased to inform you that your manuscript has been deemed suitable for publication in PLOS ONE. Congratulations! Your manuscript is now being handed over to our production team.

Kind regards, 

on behalf of

Dr. Anwar P.P. Abdul Majeed 

Academic Editor

PLOS ONE